# The Effects of COVID-19 Lockdown on the Sleep Quality of Children

**DOI:** 10.3390/children10060944

**Published:** 2023-05-26

**Authors:** Roberto Lopez-Iracheta, Laura Moreno-Galarraga, Jose Manuel Moreno-Villares, Oscar Emilio Bueso-Asfura, Miguel Angel Martinez-Gonzalez, Nerea Martin-Calvo

**Affiliations:** 1Department of Paediatrics, University Clinic of Navarra, Av. de Pío XII, 36, 31008 Pamplona, Spain; rlopezi@unav.es; 2Department of Paediatrics, University Hospital of Navarra, C/Irunlarrea sn, 31006 Pamplona, Spain; 3IdiSNa, 31008 Pamplona, Spain; 4Children’s Hospital, BCH-Harvard Medical School, Boston, MA 02115, USA; 5School of Medicine, Department of Preventive Medicine and Public Health, University de Navarra, 31009 Pamplona, Spain; 6CIBER de Fisiopatología de la Obesidad y la Nutrición, Instituto de Salud Carlos III, 28029 Madrid, Spain

**Keywords:** COVID-19 lockdown, sleep quality, parental education, SENDO project

## Abstract

Introduction: The COVID-19 lockdown has caused important changes in children’s routines, especially in terms of nutrition, physical activity, screen time, social activity, and school time. Regarding these changes, recent studies show that the COVID-19 lockdown is associated with higher levels of anxiety and depression in children. The objective of this study was to assess changes in sleep quality in Spanish children during the lockdown decreed by the Spanish government between March and June 2020. Methodology: We compared the BEAR (bedtime, excessive daytime sleepiness, awakening during the night, and regularity and duration of sleep) scores of 478 participants (median age = 7.5 years; 48% girls) in the SENDO project during the periods before, during, and after lockdown. The questionnaires were filled out by one of the parents. We used hierarchical models with two levels of clustering to account for the intra-cluster correlation between siblings. The interaction of time with a set of a priori selected variables was assessed by introducing the interaction term into the model and calculating the likelihood ratio test. Results: The mean scores in the BEAR questionnaire referred to the periods before, during, and after lockdown were 0.52 (sd 1.25), 1.43 (sd 1.99), and 1.07 (sd 1.55), respectively. These findings indicate a deterioration in sleep quality during the period of confinement. Parental level of education was found to be an effect modifier (*p* for interaction = 0.004). Children whose parents had higher education (university graduates or higher) showed a smaller worsening than those without. Conclusion: Our study shows that the COVID-19 lockdown was associated with a significant worsening of sleep quality. Moreover, although the end of the lockdown brought about a slight improvement, mean scores on the BEAR scale remained significantly higher than before the lockdown, suggesting that the consequences for sleep quality could persist over time. This worsening was higher in children whose parents had lower educational degrees. Helping children maintain healthy sleeping habits despite the circumstances and providing early psychological support when needed is important to prevent negative psycho-physical symptoms due to lockdown that could persist over the years.

## 1. Introduction

The COVID-19 pandemic changed the lives of a large part of the population worldwide. Although the hardest times of the pandemic have passed, we continue to face medium- and long-term complications and consequences for public health, the economy, and society [1]. The most important long-term consequences of the pandemic seem to be related to its psychological impact [2].

It is known that the COVID-19 quarantine caused important changes in children’s routines, especially in terms of nutrition, physical activity, screen time, social activity, and school time [3]. Regarding these changes, recent studies show that the COVID-19 lockdown was also associated with higher levels of anxiety and depression in children [3,4,5,6,7,8,9].

Confinement conditions can lead to forced inactivity and an increase in sedentary behavior, which is known to be associated with a higher risk of physical and psychological adverse conditions such as obesity, muscle atrophy, cardiovascular vulnerability [10], anxiety, and depression [11]. Additionally, fear of becoming infected and a hostile environment brought on by lockdown have been linked to higher levels of anxiety and depression [12], which are related to the quantity and quality of sleep.

Getting a good night’s sleep is necessary for correct growth and development in childhood and adolescence. Poor sleep or insufficient sleep may affect concentration, memory, and behavior [13]. One study with Italian children showed that irritability and sleep disorders were the most frequent problems referred to by children and adolescents during lockdown [9]. These sleeping problems included insomnia, difficulty falling asleep and waking up, and night awakenings.

Adolescence is a critical period in which many physical, psychological, and socio-cognitive changes occur that may have long-term implications for health and well-being. Adolescents seem to be the most vulnerable to developing sleep problems during confinement. Many studies reported significant changes in subjective well-being and an increase in anxiety symptoms in adolescents during lockdown [12,13]. One study carried out in the United States found an increase in the relative mean number of searches for information about suicide, mental health, and financial problems on the Internet [14]. 

The psychological consequences of the COVID-19 pandemic and lockdown should not be undervalued. The dramatic lifestyle changes from this pandemic posed a real threat to mental health because they raised symptoms in previously healthy people and exacerbated symptoms in people with latent disorders. These consequences must be known and studied to adequately treat current patients and to develop preventive strategies to be implemented in the event of similar catastrophes. 

Spain was one of the most affected countries by the pandemic in Europe, and its lockdown conditions were among the most restrictive. On 14 March 2020, the Spanish Government declared a state of alarm and established a lockdown (RD463/2020) that lasted almost 50 days (until 2 May 2020) and was followed by four phases of de-escalation. During that period, only healthcare professionals and people in essential services could leave their homes. The rest of the citizens, including children, could only go out to buy food and basic supplies, not for a walk or to work out. The state of alarm and lockdown ended on 21 June 2020. The main objective of this study was to assess changes in sleep quality in Spanish children during the COVID-19 lockdown. We hypothesize a worsening of sleep quality in children because of an increase in anxiety levels, an increase in screen consumption, and less time for physical activity.

## 2. Methodology

### 2.1. Study Aim, Design, and Setting

The SENDO project (“Seguimiento del Niño para un Desarrollo Óptimo”, in Spanish, [Child Follow-up for Optimal Development]; www.proyectosendo.es, accessed on 5 March 2023) is a multipurpose, ongoing, prospective pediatric cohort focused on studying the effect of diet and lifestyle on children and adolescent health. Participants are invited to enter the cohort by their pediatrician at their primary care health center or by the research team at school. Recruitment started in 2015, and it is permanently open. The inclusion criteria are (1) age between 4 and 5 years and (2) residence in Spain. The only exclusion criterion is a lack of access to an internet-connected device to complete the questionnaires. Further information on this cohort study design has previously been reported in detail elsewhere [15].

Information is collected at baseline and updated every year through online self-administered questionnaires completed by parents. In September 2020, an additional brief questionnaire was sent to participants to collect information on lifestyle changes experienced during the Spanish lockdown between March and June 2020. Answers were collected until 31 December 2020.

### 2.2. Participants

Among the 832 participants recruited before September 2020, 485 completed the additional questionnaire regarding changes in lifestyle during lockdown (participation proportion: 58%). Participants with missing information in four or more questions regarding sleeping quality were excluded from the analysis (n = 7). Participants with fewer missing items were contacted to complete the information. The final sample consisted of 478 participants.

### 2.3. Assessment of the Outcome and Covariates

The baseline questionnaire of the SENDO project collects information on sociodemographic, lifestyle, and diet-related variables. The information referred to as lockdown was collected through a brief additional questionnaire passed between September and December 2020. This questionnaire included questions related to changes in sleeping habits (the BEAR questionnaire), dietary habits (changes in food consumption), physical activities (type and time), and sedentary time (including screen time).

Sleep habits were assessed with the BEAR questionnaire. The BEAR questionnaire is one of the most commonly used tools to identify sleep problems in children and has already been validated in Spain [16,17,18,19]. BEAR is the acronym for B—bedtime, E—excessive daytime sleepiness, A—awakening during the night, and R—regularity and duration of sleep. Really, the original tool is BEARs, which include snoring. We did not take snoring into account because, in most children, the cause of snoring is the anatomy of the mouth, head, and neck, particularly when they have enlarged adenoids or tonsils. The environment seems to be less important in that regard. Moreover, most of these children sleep alone, and sometimes parents do not realize if they snore or not. We use this user-friendly screening tool that helps identify sleeping problems in children and includes the questions: (1) Are you reluctant to go to bed? (2) Do you have any difficulty falling asleep? (3) Do you usually wake up more than two times per night? (4) If you wake up at night, do you have any difficulty falling back asleep? We assigned points to each answer as follows: always/very often (3 points), often (2 points), occasionally (1 point), or almost never/never (0 points). Each participant completed the BEAR questionnaire three times (including information for periods before, during—March to June 2020—and after lockdown). We calculated the total score in each questionnaire by adding the punctuation in the four questions. Final scores ranged between 0 and 12, with higher scores meaning more sleeping problems. We obtained three total scores in the BEAR questionnaire for each participant referring to periods before, during, and after lockdown. For descriptive purposes, participants were divided into two groups based on their basal score on the BEAR questionnaire, referring to the period before lockdown (0 vs. ≥1 point).

### 2.4. Statistical Analysis

Participant characteristics are described by the basal BEAR score (BEAR score before lockdown: 0 vs. ≥1). We used mean values and standard deviations for quantitative variables and percentages for categorical ones. Between-group comparisons were made using the Student *t*-test for quantitative variables and χ^2^ test for qualitative ones. We used McNemar’s test to assess the difference between paired proportions.

We compared scores on the BEAR questionnaire for the three periods (before, during, and after lockdown) using repeated measures for each participant. We used hierarchical models with two levels of clustering to account for the intra-cluster correlation between siblings. In further analyses, participants were divided into two groups depending on parental educational level (university graduates or above).

Interaction between time and a priori selected variables (i.e., sex, number of siblings, number of cohabitants, parental education, having a pet, having a mobile phone, physical activity (METS), BMI (kg/m^2^), moderate or intense activity (hours/week), and adherence to the Mediterranean diet) was assessed by introducing the interaction term into the model and calculating a likelihood ratio test.

Analyses were carried out using the software STATA 15.0 (Stata Corporation, College Station, TX, USA). All *p*-values were two-tails. Statistical significance was determined at the conventional cut-off point of *p* < 0.05.

### 2.5. Ethical Considerations

The SENDO project follows the rules of Declaration of Helsinki on the ethical principles for medical research in human beings. This study was approved by the Ethics Committee for Clinical Research of Navarra (Pyto 2016/122). Informed consent was obtained from all participants’ parents at recruitment.

## 3. Results

This study includes 479 participants (226 girls) with a mean age of 7.5 years (sd: 1.8), all of whom were recruited from the SENDO cohort. 371 children scored 0 points in the basal BEAR questionnaire (the period before lockdown). This means that they did not present any previous sleeping disorders. The mean score on the basal BEAR questionnaire in patients with previous sleep problems was 0.52 (sd: 1.24). The mean sleep time before confinement was 9.2 h (sd: 0.44).

The basal characteristics of participants as determined by the BEAR questionnaire are shown in Table 1. We observed no differences between groups in the sociodemographic and lifestyle characteristics of children or their family characteristics.

Figure 1 shows the responses to each question in the BEAR questionnaire. For each question, the percentage of responses before, during, and after lockdown was represented. The most frequent category of response in all the questions was “almost never/never” either before, during, or after lockdown. However, the proportion of participants who answered “never/almost never” decreased during the lockdown and increased after it without reaching the initial level.

Mean scores in the BEAR questionnaire referring to the periods before, during, and after lockdown were 0.52 (sd 1.25), 1.43 (sd 1.99), and 1.07 (sd 1.55), respectively (Table 2). Children with previous sleeping disorders (≥1 point in the basal BEAR questionnaire) had significantly higher mean scores in the BEAR questionnaires during (3.11 vs. 0.94 points) and after (2.74 vs. 0.59 points) lockdown. We observed a similar trend in these children without any previous sleep problems (0 points in the basal BEAR questionnaire). In conclusion, we found that the mean score in the BEAR questionnaire significantly increased during lockdown (worse sleep quality) and significantly decreased after it (better sleep quality). However, it did not reach the initial level, and the mean score in the BEAR questionnaire referring to the period after lockdown was significantly higher (worse sleep quality) than before. That trend was observed in the whole sample and each group.

Additionally, the mean leisure screen time of participants was 1.13 h/day (sd: 0.81) before the lockdown and 2.65 h/day (sd: 1.69) during it (*p* < 0.001). The mean time of moderate or vigorous physical activity was 1.27 h/day (sd: 0.99) before the lockdown and 0.79 h/day (sd: 0.96) during (*p* < 0.001).

Parental level of education was found to be an effect modifier (*p* for interaction = 0.004). Although similar trends in the mean score were observed in both groups (an increase in the mean score during lockdown and a decrease after it), children whose parents had higher education (university graduates or higher) showed a smaller increase in the comparison before vs. during lockdown (0.86 vs. 1.17 points) and a higher decrease in the comparison during vs. after it (0.38 vs. 0.2 points). Children whose parents did not have high education did not present a significant improvement in sleep quality after the lockdown according to their mean score in the BEAR questionnaire (1.75 vs. 1.55 points) (Table 3). The comparison of mean scores in the questionnaire before lockdown showed no difference between groups. However, children whose parents did not have higher education showed significantly higher mean scores in the questionnaire, referring to both the period during (difference = 0.38 points) and after the (difference = 0.56 points) lockdown (Table 3).

## 4. Discussion

In this study of 479 children from the SENDO project, we found that the lockdown decreed in Spain between March and June 2020 as a consequence of the COVID-19 pandemic was associated with a significant worsening of sleep quality. This effect was particularly pronounced among children whose parents did not have higher education. Furthermore, we found that although it improved after the lockdown, the sleep quality did not reach previous levels, suggesting the consequences of the lockdown on sleep quality could have a long-term effect. The increase in anxiety disorders in children, together with the increase in screen time that persisted after the end of the lockdown, may explain the worsening of children’s sleep quality. A study [20] with 715 minors from the United Kingdom found that not only did they take longer to fall asleep, but also that for every daily hour they spent in front of a screen, nighttime sleep was reduced by 26 min. A review of more than twenty previous studies [21] that included more than 125,000 minors pointed out that children who used screens before going to sleep had double the risk of insufficient sleep time than those who did not. This increased risk can be attributed to the fact that exposure to bright light from screens suppresses the production of melatonin, which is the hormone responsible for inducing sleep.

To our knowledge, this is the first study in Spain that shows a long-term impact of the COVID-19 lockdown on children’s quality of sleep. Our results highlight the importance of assessing sleep quality in children and developing preventive measures for future possible lockdowns. Our results agree with previously published articles reporting that the COVID-19 pandemic has increased levels of stress in children and their relatives [22], as well as sleep disturbances. A recent meta-analysis showed that the pooled prevalence of any sleep disturbance in children during the pandemic was 54% (95% CI: 50–57%) [23]. One cross-sectional study carried out in China with 53,730 participants found that more than 35% of participants developed psychological distress [24]. Additionally, a similar study also performed in China with 1304 participants showed that more than 50% presented negative emotional influences during the COVID-19 pandemic [25]. Those findings are consistent with previous knowledge that suggested that long periods of social isolation negatively impact people’s psychological well-being [26], leading to depression, stress, and anxiety [27]. We hypothesize that the worsening of sleep quality in Spanish children observed in this study could be at least partially explained by an increase in children’s anxiety levels.

Recent studies found that social isolation during the COVID-19 pandemic greatly reduced the level of physical activity in both male and female students [28,29], which may have worsened their sleep quality. The results of one study among 516 parents to collect data about 860 children and adolescents (49.2% girls) aged between 3 and 16 years suggested that COVID-19 confinement reduced physical activity levels, increased both screen exposure and sleep time and reduced fruit and vegetable consumption [30]. Physical activity can promote a positive mood [31], help maintain a healthy weight, and establish self-esteem both in children and adolescents [32]. Vigorous physical activity can significantly reduce adolescents’ stress, anxiety, and depression [33] and protect them from metabolic syndrome [34]. The use of electronic devices has been associated with worse sleep quality [35,36], and their use increased during the lockdown [37].

We found that parental education was an effect modifier. In our study, children whose parents had a low or middle level of education showed a greater worsening of sleep quality. This finding is not surprising considering recent publications that showed that the COVID-19 lockdown had an important impact on parents and caregivers who developed high levels of anxiety, stress, and depression [38]. That impact seemed to be greater in mothers [39], especially in those with lower levels of education or who were unemployed [40]. Therefore, we hypothesize that children of parents with medium or low levels of education showed a greater deterioration of sleep quality either because the level of anxiety in their homes was higher or because those parents had fewer resources to help their children sleep.

The COVID-19 pandemic and subsequent lockdown led to an alarming increase in levels of anxiety, depression, irritability, boredom, inattention, and sleep problems in children [41]. These consequences are not exclusive to a pandemic and can appear in the event of any other natural or man-made disaster. Public health strategies are needed to prevent this type of consequence through a multidisciplinary approach that helps children better manage and control stress and anxiety [42].

Despite our results, this study had some limitations. Self-reported information is susceptible to misclassification bias. Since the SENDO project information is updated every year through online questionnaires, families are used to this method of data collection, which reduces the risk of systematic error.

In addition, participants in the SENDO project are mostly white children from medium- to high-level educated families. Although this may hamper the generalizability of our results, it increases the validity of the responses and reduces confounding by socioeconomic status. Due to the observational design, we cannot deny the possibility of residual confounding by variables that were not accounted for.

## 5. Conclusions

In conclusion, many studies have associated the COVID-19 pandemic and lockdown with psychological and lifestyle changes. Our study shows that the COVID-19 lockdown is also associated with a significant worsening of sleep quality. Moreover, although the end of the lockdown brought about a slight improvement, mean scores on the BEAR scale remained significantly higher than before. This suggests that the consequences for sleep quality could persist over time. This worsening is higher in children whose parents did not have higher education. Helping children maintain healthy sleeping habits despite the circumstances and providing early psychological support when needed is important to prevent negative psycho–physical symptoms from lockdown that could persist over the years. Encourage children to engage in physical activity outdoors whenever possible. This should be considered on future occasions. These recommendations should be considered in order to develop preventive strategies to be implemented in the event of similar catastrophes. More studies are needed to determine the long-term effects of the COVID-19 lockdown, mitigate them, and, if necessary, define prevention strategies that could be implemented in the event of a similar situation occurring again.

## Figures and Tables

**Figure 1 children-10-00944-f001:**
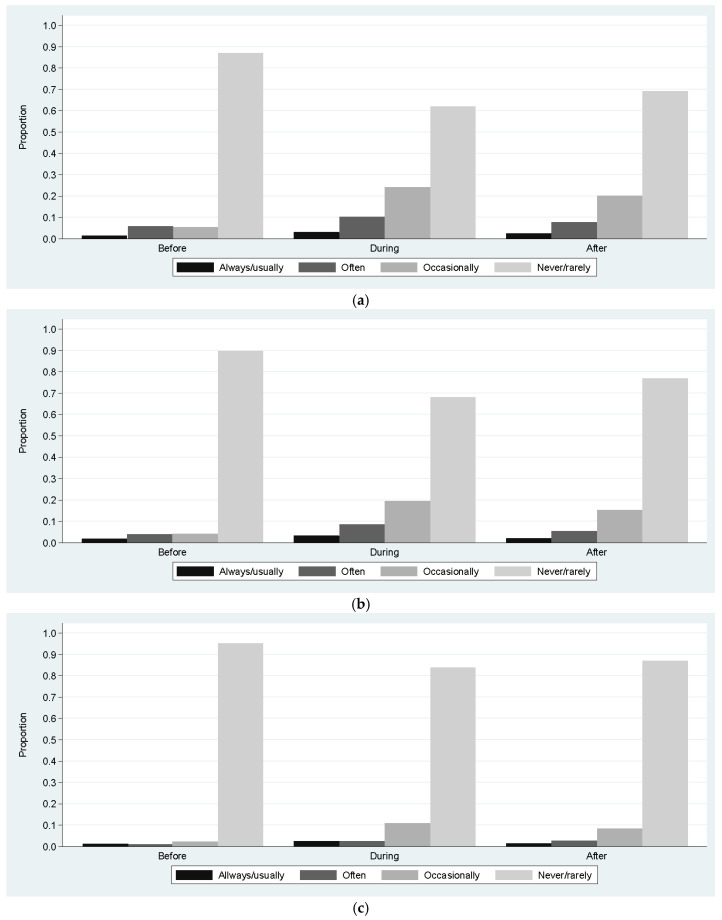
Description of the score of quality of sleep. (**a**) Proportion of responses to the question Are you reluctant to go to bed? The proportion of participants who answered “never/almost never” decreased during lockdown (*p* < 0.05) and increased after it (*p* < 0.05). (**b**) Proportion of responses to the question Do you have any difficulty falling asleep? The proportion of participants who answered “never/almost never” decreased during lockdown (*p* < 0.05) and increased after it (*p* < 0.05). (**c**) Proportion of responses to the question Do you usually wake up more than two times per night? The proportion of participants who answered “never/almost never” decreased during lockdown (*p* < 0.05) and increased after it (*p* < 0.05). (**d**) Proportion of responses to the question If you wake up at night, do you have any difficulty falling back asleep? The proportion of participants who answered “never/almost never” decreased during lockdown (*p* < 0.05) and there were no statistical differences after it (*p* = 0.11).

**Table 1 children-10-00944-t001:** Characteristics of participants in the SENDO project by basal BEAR score (before lockdown). Numbers are in percent or mean (standard deviation).

	Baseline Scores in the BEAR Questionnaire		
	0 Points	≥1 Points	*p*-Value
**N**	**371**	**107**	
Sex (%girls)	46.3	50.9	0.399
Age (years)	7.4 (1.8)	7.5 (1.8)	0.614
BMI (kg/m^2^)	16.2 (2.0)	16.0 (1.7)	0.501
Moderate or intense physical activity (hours/week)	9.1 (7.3)	8.0 (5.2)	0.153
Leisure screen time (hours/day)	1.1 (0.8)	1.1 (0.9)	0.677
Adherence to the Mediterranean diet (score in the KIDMED index)	6.0 (1.9)	6.0 (1.7)	0.910
Number of siblings	1.4 (1.1)	1.3 (1.0)	0.497
Number of cohabitants	3.6 (1.3)	3.9 (4.2)	0.250
Exposure to passive smoking (%)	5.8	9.3	0.428
Pet owners (%)	26.1	19.6	0.168
Mobile phone owners (%)	1.1	0.9	0.898
Mother’s age (years)	42.4 (4.2)	42.0 (4.4)	0.436
Father’s age (years)	43.3 (5.6)	43.4 (4.3)	0.958
Mother/Father superior studies (university graduate or above)	86.0	82.2	0.338

**Table 2 children-10-00944-t002:** Mean (standard deviation) scores of the BEAR questionnaire referred to the periods before, during, and after lockdown in the whole sample and by groups depending on the participant’s score in the questionnaire referred to the period before lockdown.

	Before the Lockdown	During Lockdown	After Lockdown
Mean score (the whole sample)	0.52 (1.25)	1.43 (1.99) ^a^	1.07 (1.55) ^b,c^
Mean score in children with no previous sleep problems	0 (0)	0.94 (1.54) ^d^	0.59 (1.06) ^e,f^
Mean score in children with previous sleep problems	2.33 (1.66)	3.11 (2.42) ^g^	2.74 (1.83) ^h,i^

^a^ during vs. before: *p* < 0.001, ^b^ after vs. before: *p* < 0.001, ^c^ after vs. during: *p* < 0.001. ^d^ during vs. before: *p* < 0.001, ^e^ after vs. before: *p* < 0.001, ^f^ after vs. during: *p* < 0.001. ^g^ during vs. before: *p* < 0.001, ^h^ after vs. before: *p* = 0.009, ^i^ after vs. during: *p* = 0.017.

**Table 3 children-10-00944-t003:** Mean (standard deviation) score of the BEAR questionnaire referred to the periods before, during, and after lockdown in the whole sample and by groups depending on the participant’s parents’ level of education.

	Before Lockdown	During Lockdown	After Lockdown
Mean score in the whole sample	0.52	1.43 ^a^	1.07 ^b,c^
Mean score in children whose parents did not have higher education	0.58 (1.24)	1.75 (2.30) ^d^	1.55 (1.92) ^e,f^
Mean score in children whose parents had higher education	0.51 (1.25)	1.37 (1.93) ^g^	0.99 (1.46) ^h,i^

^a^ during vs. before: *p* < 0.001, ^b^ after vs. before: *p* < 0.001, ^c^ after vs. during: *p* < 0.001. ^d^ during vs. before: *p* < 0.001, ^e^ after vs. before: *p* < 0.001, ^f^ after vs. during: *p* = 0.311. ^g^ during vs. before: *p* < 0.001, ^h^ after vs. before: *p* < 0.001, ^i^ after vs. during: *p* < 0.001

## Data Availability

The data presented in this study are available on request from the corresponding author. The data are not publicly available due to privacy reasons.

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
