# Peer review of "The Effects of COVID-19 Lockdown on the Sleep Quality of Children"

_children, 2023, doi:10.3390/children10060944_

Round 1

Reviewer 1 Report

Thank you for the opportunity to review the article "The effects of COVID-19 lockdown on children's sleep quality". 

The article refers to the negative consequences of the outbreak of the coronavirus with a focus on the quality of children's sleep. The article adds and expands the knowledge that already exists in the literature about the effect of the coronavirus in all kinds of countries and cultures.

The article has positive points and strengths. However, I have comments and suggestions.

1) Abtract: The Authors mentioned the abbreviation "BEAR" but did not mention what it means. Authors should provide the full name and then the abbreviation (page 1, line 12).

2) The age of the children included in the study should be provided in the Abstract.  There is room to indicate whether the respondents were the children or the parents.

3) I suggest rewriting the conclusions in the abstract, it is more suitable for the findings.

4) Introduction: Authors stated "Regarding these changes, recent studies show that Covid-19 lockdown was also associated with higher levels of anxiety and depression in children" (lines 34-35) and they cited only two studies. I think there is room to add more references on this topic. See (Francisco et al., 2020; Ghanamah & Eghbaria-Ghanamah, 2021; 2023; Giménez-Dasí et al., 2020; Jiao et al., 2020; Yeasmin et al., 2020).

5) The research questions and/or hypotheses are not clearly provided.

6) Method: What analysis did the authors use to test for differences in qualitative variables? How did the authors deal with repeated testing (potential increase in Type I error)?

7) Results: The Figures should be clearer, it should be provided in the figures (with signs) what was significant.  The tables are hard to follow. I suggest providing a figure with the results of the hierarchical models.

Overall, there is room to improve the manuscript. I seriously debated whether to reject the article or to suggest for major revision.  I decided to give the authors a chance to revise it. I hope that by giving the authors the chance to revise the manuscript they can respond to the comments and suggestions. 

Author Response

Thank you for the opportunity to review the article "The Effects of COVID-19 Lockdown on children's sleep quality". 

The article refers to the negative consequences of the outbreak of the coronavirus with a focus on the quality of children's sleep. The article adds and expands the knowledge that already exists in the literature about the effect of the coronavirus in all kinds of countries and cultures.

The article has positive points and strengths. However, I have comments and suggestions.

  • Abstract: The Authors mentioned the abbreviation "BEAR" but did not mention what it means. Authors should provide the full name and then the abbreviation (page 1, line 12).

Response

 BEAR is a user-friendly screening tool to help identify sleep problems in children. Is the acronym of B - Bedtime, E - Excessive Daytime Sleepiness, A - Awakening During the Night, and R - Regularity and Duration of Sleep. Really, the original tool is BEARS that

  • The age of the children included in the study should be provided in the Abstract.  There is room to indicate whether the respondents were the children or the parents.

Response

We add this. The mean age is 7.5 years. Th questionnaires were filled by on od the parents.

  • I suggest rewriting the conclusions in the abstract, it is more suitable for the findings.

Response

Done. We have rewritten the paragraph:

 Our study shows that Covid-19 lockdown was associated with a significant worsening of sleep quality. Moreover, although the end of lockdown brought about a slight improvement, mean scores in the BEAR scale remained significantly higher than before the lockdown, suggesting that the consequences on sleep quality could persist overtime. This worsening was higher in children whose parents had lower education degrees. Helping children to maintain healthy sleeping habits despite the circumstances, and providing early psychological support when needed is important to prevent negative psycho-physical symptoms due to lockdown that could persist over the years

  • Introduction: Authors stated "Regarding these changes, recent studies show that Covid-19 lockdown was also associated with higher levels of anxiety and depression in children" (lines 34-35) and they cited only two studies. I think there is room to add more references on this topic. See (Francisco et al., 2020; Ghanamah & Eghbaria-Ghanamah, 2021; 2023; Giménez-Dasí et al., 2020; Jiao et al., 2020; Yeasmin et al., 2020).

Response

Thank you for your advice. We have added the following references:

Francisco R, Pedro M, Delvecchio E, Espada JP, Morales A, Mazzeschi C, Orgilés M. Psychological Symptoms and Behavioral Changes in Children and Adolescents During the Early Phase of COVID-19 Quarantine in Three European Countries. Front Psychiatry. 2020 Dec 3;11:570164.

Ghanamah R, Eghbaria-Ghanamah H. The Psychological Effects of Coronavirus on Children in the Perception of Arab Israeli Parents Sample The Psychological Effects of Coronavirus on Children in the Perception of Arab Israeli Parents Sample. Child & Youth Services. 2023.

Giménez-Dasí M, Quintanilla L, Lucas-Molina B, Sarmento-Henrique R. Six Weeks of Confinement: Psychological Effects on a Sample of Children in Early Childhood and Primary Education. Front. Psychol., 08 October 2020. Volume 11 – 2020.

Jiao WY, Wang LN, Liu J, Fang SF, Jiao FY, Pettoello-Mantovani M, Somekh E. Behavioral and Emotional Disorders in Children during the COVID-19 Epidemic. J Pediatr. 2020 Jun;221:264-266.e1.

Yeasmin S, Banik R, Hossain S, Hossain MN, Mahumud R, Salma N, Hossain MM. Impact of COVID-19 pandemic on the mental health of children in Bangladesh: A cross-sectional study. Child Youth Serv Rev. 2020 Oct;117:105277.

  • The research questions and/or hypotheses are not clearly provided.

Response

Thank you for your comment. We have added the following sentence to the introduction:

The main objective of this study was to assess changes in sleep quality of Spanish children during the Covid-19 lockdown. We hypotheses a worsening of sleep quality in children because of an increase in anxiety levels, an increase in screen consumption and less time for physical activity.

  • Method: What analysis did the authors use to test for differences in qualitative variables? How did the authors deal with repeated testing (potential increase in Type I error)?

Response

Thanks. We have tried to clarify this issue with the following sentence:

Between-groups comparisons were made using Student t-test for quantitative variables and χ2 tests for qualitative ones. We used McNemar's test, to assess the difference between paired proportions.

  • Results: The Figures should be clearer, it should be provided in the figures (with signs) what was significant.  The tables are hard to follow. I suggest providing a figure with the results of the hierarchical models.

Response

Thank you. We try to clarify the tables and figures and we have added the p-value for the difference in proportions for each question in the figure 1.

Overall, there is room to improve the manuscript. I seriously debated whether to reject the article or to suggest for major revision.  I decided to give the authors a chance to revise it. I hope that by giving the authors the chance to revise the manuscript they can respond to the comments and suggestions. 

We really appreciate this comment from you. We have tried to work hard to make this research as a part of the Ph Thesis from one of the authors (RLI). We hope you have found suitable our answers and corrections.

Reviewer 2 Report

The study describes the sleep quality in Spanish children before, during and after the Covid-19 lockdown. A nice study. The writing per se is good, although the authors are requested to address some issues to improve clarity, especially the time reference frame of the epoch of data collected in the children with a mean age of 7.5 y. The issues are listed below.

·      In the Abstract methodology: It is crucial to include a statement about the longitudinal aspect of the SENDO project, i.e., the data collected in children (between March and June 2020) represented an epoch of data within a longitudinal study.

·      In the Results section: Clarification is required as to whether the current data presented of children with a mean age of 7.5y represented a follow-up of the same children enrolled in 2015.

·      The acronym of the sleep screening tool in children is BEARS, not BEAR. It is not clear why the last question on “snoring” was dropped (see Lines 106-108).

·      The authors used the term “sleep disorders” yet “snoring” has not been evaluated in the BEARS questionnaire. Clarification on this is required.

·      Clarity is sought regarding two questionnaires that were added. Were they the same?

o  A questionnaire on lifestyle data added in September 2020 and a questionnaire on sleep etc added in September and December 2020.

·     The authors mentioned “anxiety” (Line 212) and “stress” (Line 226) in the Discussion. Please clarify if and how anxiety and stress data were collected, and a brief description to be included in the Methods.

 Specific comments

Line 87: Please clarify what lifestyle data were collected in the added questionnaire.

Line 15: Please report the Mean age and SD, and sex and proportions.

Line 60: Suggest replace "depreciated" with "undervalued"

Line 94: Please add "equal" symbol for n=7.

Line 98-102: This paragraph is causing some confusion and the info presented is repetitive. Please move this paragraph and incorporate it into Line 86-89, if indeed they are the same questionnaire.

Line 111: Please specify when the 3 timepoints were (over March to June 2020). Was "during" around the middle of the collection period?

Line 142: Inclusion criteria for the study stated children were between 4 and 5 years old. The children in question had a mean age of 7.5y. Please confirm if these were the same children enrolled into the study back in 2015?

Line 144-145: Please clarify why "sleep disorders", given that the authors had left out the last question on snoring in the BEARS questionnaire.

Line 170: please change "previous sleeping disorders" to "sleep problems" as reported correctly in Table 2.

Line 212:  Please clarify how anxiety data were captured? Please report this in the Methods.

Line 224: It would be helpful if authors could list some future preventive measures.

Line 226: How was stress captured. Please specify. Whilst stress may be a factor that deters sleep, other factors like dietary intake and light exposure etc should also be considered.

Author Response

The study describes the sleep quality in Spanish children before, during and after the Covid-19 lockdown. A nice study. The writing per se is good, although the authors are requested to address some issues to improve clarity, especially the time reference frame of the epoch of data collected in the children with a mean age of 7.5 y. The issues are listed below.

  • In the Abstract methodology: It is crucial to include a statement about the longitudinal aspect of the SENDO project, i.e., the data collected in children (between March and June 2020) represented an epoch of data within a longitudinal study.
  • In the Results section: Clarification is required as to whether the current data presented of children with a mean age of 7.5y represented a follow-up of the same children enrolled in 2015.
  • The acronym of the sleep screening tool in children is BEARS, not BEAR. It is not clear why the last question on “snoring” was dropped (see Lines 106-108).
  • The authors used the term “sleep disorders” yet “snoring” has not been evaluated in the BEARS questionnaire. Clarification on this is required.
  • Clarity is sought regarding two questionnaires that were added. Were they the same?

o  A questionnaire on lifestyle data added in September 2020 and a questionnaire on sleep etc added in September and December 2020.

  • The authors mentioned “anxiety” (Line 212) and “stress” (Line 226) in the Discussion. Please clarify if and how anxiety and stress data were collected, and a brief description to be included in the Methods.

 Specific comments

Line 87: Please clarify what lifestyle data were collected in the added questionnaire.

Line 15: Please report the Mean age and SD, and sex and proportions.

Line 60: Suggest replace "depreciated" with "undervalued"

Line 94: Please add "equal" symbol for n=7.

Line 98-102: This paragraph is causing some confusion and the info presented is repetitive. Please move this paragraph and incorporate it into Line 86-89, if indeed they are the same questionnaire.

Line 111: Please specify when the 3 timepoints were (over March to June 2020). Was "during" around the middle of the collection period?

Line 142: Inclusion criteria for the study stated children were between 4 and 5 years old. The children in question had a mean age of 7.5y. Please confirm if these were the same children enrolled in the study back in 2015.

Line 144-145: Please clarify why "sleep disorders", given that the authors had left out the last question on snoring in the BEARS questionnaire.

Line 170: please change "previous sleeping disorders" to "sleep problems" as reported correctly in Table 2.

Line 212:  Please clarify how anxiety data were captured. Please report this in the Methods.

Line 224: It would be helpful if authors could list some future preventive measures.

Line 226: How was stress captured? Please specify. Whilst stress may be a factor that deters sleep, other factors like dietary intake and light exposure, etc should also be considered.

Response

Thank you for your comments. Really, the original tool is BEARS that include snoring. We did not take account of snoring because in most children the cause of snoring is because of the anatomy of mouth, head and neck anatomy, particularly when they have enlarged adenoids or tonsils. The environment seems to be less important in that. Moreover, most of these children sleep alone and sometimes parents do not realize if they snore or not.

We therefore respond to each specific comment below:

  • Line 87: done. This questionnaire included questions related to changes in sleeping habits (BEAR questionnaire), dietary habits (changes in food consumption), physical activities (type and time), and sedentary time (including screen time). Now lines 103-106
  • Line 15: done thank you. Now line 10
  • Line 60: done, thank you. Now, line 62
  • Line 94: done, thank you. Line 98
  • Line 98-102: We have tried to clarify this paragraph (now line 101-106)

The baseline questionnaire of the SENDO project collects information on sociodemographic, lifestyle and diet-related variables. The information referred to lockdown was collected through a brief additional questionnaire passed between September and December 2020. This questionnaire included questions related to changes in sleeping habits (BEAR questionnaire), dietary habits (changes in food consumption), physical activities (type and time), and sedentary time (including screen time).

  • Line 111: ok, I add this information (line 121)
  • Line 142: All patients were part of the SENDO cohort. They have between 4-5 years at the beginning of the project, and currently 7.5 as a mean, at the time of this study, We have added a sentence to clarify (lines 153-154)
  • Line 144-145: explained in methodology, thank you (lines 107-114)
  • Line 170: changed, thank you. (line 156)
  • Line 212 and line 226: Thank you for your comment

We did not assess anxiety or stress in our study, so these data were not from our study. We speculate, according to the literature, that both factors could contribute to sleep disturbances.

  • Line 224: Thank you for your comment. We have added the following sentence:

Helping children to maintain healthy sleeping habits despite the circumstances and providing early psychological support when needed is important to prevent negative psycho-physical symptoms from lockdown that could persist over the years. Let children do physical activity outside if it is possible should be considered in future occasions. These recommendations should be considered in order to develop preventive strategies to be implemented in the event of similar catastrophes.(lines 257-265)

Reviewer 3 Report

this study described children's subjective sleep outcomes during the COVID-19 lockdown (n=478) and reported an increase in sleep issues during this period.

Overall I have concerns with the added value of this study. Several studies were already published on this topic. Systematic review and meta-analysis were even conducted and have already provided us with all the necessary information.

Author Response

this study described children's subjective sleep outcomes during the COVID-19 lockdown (n=478) and reported an increase in sleep issues during this period.

Overall I have concerns with the added value of this study. Several studies were already published on this topic. Systematic review and meta-analysis were even conducted and have already provided us with all the necessary information.

Response

Thank you for your comment. To our knowledge, there are few articles about the effect of the lockdown on sleep quality  and most of them are in adult. There are no data on Spanish population at this age.

Reviewer 4 Report

"The effects of COVID-19 lockdown on the sleep quality of children"

- has to be edited by a native speaker

- design & methods can be improved - more details

- conclusion is missing

Author Response

"The effects of COVID-19 lockdown on the sleep quality of children"

- has to be edited by a native speaker

- design & methods can be improved - more details

- conclusion is missing

Response

  • Thank you. We have the paper reviewed by a native speaking English.
  • As in the answer to previous reviews we have added some considerations on the methodology as well as the design. To answer you more correctly: SENDO project is a longitudinal study based on a cohort of Spanish children, from different parts of the country, who were 4-5 yo when included. This cohort is followed by means of questionnaires at different ages until adolescence. In this study we have added an specific questionnaire to evaluate sleep quality and how the lockdown due to the COVID pandemic have modified this subject. Some additional questions were added in order to relate these changes to changes in life style, mainly physical activity as well as time in front of screens.
  • Thank you for your comments. We have added the following conclusions:

Our study shows that Covid-19 lockdown is associated with a significant worsening of sleep quality. Moreover, although the end of lockdown brought about a slight improvement, mean scores in the BEAR scale remained significantly higher than before, suggesting the consequences on sleep quality could persist overtime. This worsening is higher in children whose parents did not have higher studies. Helping children to maintain healthy sleeping habits despite the circumstances and providing early psychological support when needed is important to prevent negative psycho-physical symptoms from lockdown that could persist over the years.

Round 2

Reviewer 1 Report

well done. The authors of this publication have considered most of the remarks that I made during my first review process. There are still left some suggestions but they could be corrected quite easily.

1) Abstract; BEAR (B - Bedtime, E - Excessive Daytime Sleepiness, A - Awakening During the Night, and R - Regularity and Duration of Sleep) can be replaced by BEAR ( Bedtime, Excessive daytime sleepiness,  Awakening during the night, and  Regularity and duration of sleep). (at the discretion of the  authors and the editor)

2) Abstract: with a mean age of 7.5 years (SD 1.8) (48% girls) can be replaced by (Mage = 7.5; 48% girls). 

3) Introduction: The Authors stated that they have added the following references (line 45 ........., but I didn't find them in the revised manuscript and in the references list. 

4) I think that Table 2 should be revised and provided in a more simple and clear way. 

Author Response

Well done. The authors of this publication have considered most of the remarks that I made during my first review process. There are still left some suggestions but they could be corrected quite easily.

Response:

Thank you very much for your comments and suggestions.

1) Abstract; BEAR (B - Bedtime, E - Excessive Daytime Sleepiness, A - Awakening During the Night, and R - Regularity and Duration of Sleep) can be replaced by BEAR ( Bedtime, Excessive daytime sleepiness,  Awakening during the night, and  Regularity and duration of sleep). (at the discretion of the  authors and the editor)

Response:

We have changed the acronymous according to your suggestion

2) Abstract: with a mean age of 7.5 years (SD 1.8) (48% girls) can be replaced by (Mage = 7.5; 48% girls). 

Response:

Thanks. It has been changed in the original

3) Introduction: The Authors stated that they have added the following references (line 45 ........., but I didn't find them in the revised manuscript and in the references list. 

Response:

We added the references and in the last manuscript in .doc but it didn’t appear in th pdf. We have asked to include them.

4) I think that Table 2 should be revised and provided in a more simple and clear way. 

Response:

We have tried to clarify the table. We have simplify and explained it in the text of the manuscript. Thank you.

Reviewer 3 Report

Please note the following references

Sharma, M., Aggarwal, S., Madaan, P., Saini, L., & Bhutani, M. (2021). Impact of COVID-19 pandemic on sleep in children and adolescents: a systematic review and meta-analysis. Sleep medicine84, 259-267.

López-Bueno, R., López-Sánchez, G. F., Casajús, J. A., Calatayud, J., Gil-Salmerón, A., Grabovac, I., ... & Smith, L. (2020). Health-related behaviors among school-aged children and adolescents during the Spanish Covid-19 confinement. Frontiers in pediatrics8, 573.

Author Response

Thank you very much for these references, that have been added to the text.

The first study is very interesting but, as they say, there are fewer studies about the impact of Covid-19 lockdown and sleep quality of children: “The index study is the first systematic review capturing sleep in children during the COVID-19 pandemic. However, very few studies were eligible for inclusion and most of the included studies were not high quality…Besides, the included studies were conducted in different sample populations and had significant heterogeneity making the generalizability of this review debatable. Also, this review did not assess the published literature on the sleep of children with COVID-19 or other medical illnesses during the pandemic. These children are expected to have a higher burden of sleep problems”.

The second one is also very interesting because study many health-related behaviors that have been affected by the Covid-19 lockdown but, about sleep, it only study sleep time, but no sleep quality.